# Preservation of Biomarkers Associated with Alzheimer’s Disease (Amyloid Peptides 1-38, 1-40, 1-42, Tau Protein, Beclin 1) in the Blood of Neonates after Perinatal Asphyxia

**DOI:** 10.3390/ijms241713292

**Published:** 2023-08-27

**Authors:** Agata Tarkowska, Wanda Furmaga-Jabłońska, Jacek Bogucki, Janusz Kocki, Ryszard Pluta

**Affiliations:** 1Department of Neonate and Infant Pathology, Medical University of Lublin, 20-093 Lublin, Poland; agatatarkowska@umlub.pl (A.T.); wm.jablonska@gmail.com (W.F.-J.); 2Department of Organic Chemistry, Faculty of Pharmacy, Medical University of Lublin, 20-093 Lublin, Poland; jacekbogucki@wp.pl; 3Department of Clinical Genetics, Medical University of Lublin, 20-080 Lublin, Poland; januszkocki@umlub.pl; 4Department of Pathophysiology, Medical University of Lublin, 20-090 Lublin, Poland

**Keywords:** perinatal asphyxia, Alzheimer’s disease, amyloid, tau protein, beclin 1, autophagy, blood, biomarkers

## Abstract

Perinatal asphyxia is a complex disease involving massive death of brain cells in full-term newborns. The most impressive consequence of perinatal asphyxia is a neurodegenerative brain injury called hypoxic–ischemic encephalopathy. Management of newborns after perinatal asphyxia is very difficult due to the lack of measurable biomarkers that would be able to assess the severity of the brain injury in the future, help in the selection of therapy, assess the results of treatment and determine the prognosis for the future. Thus, these limitations make long-term neurodevelopmental outcomes unpredictable during life. Quantifying biomarkers that can detect subclinical changes at a stage where routine brain monitoring or imaging is still mute would be a major advance in the care of neonates with brain neurodegeneration after asphyxia. Understanding the effect of perinatal asphyxia on changes in blood neurodegenerative biomarkers over time, which would be commonly used to assess the severity of postpartum encephalopathy, would be an important step in developing precision in predicting the consequences of brain injuries. We urgently need more accurate early predictive markers to guide clinicians when to use neuroprotective therapy. The needed neurodegenerative biomarkers may represent neuronal pathological changes that can be recognized by new technologies such as genomic and proteomic. Nevertheless, the simultaneous blood tau protein and various amyloid changes with the addition of an autophagy marker beclin 1 after perinatal asphyxia have not been studied. We decided to evaluate serum biomarkers of neuronal injury characteristic for Alzheimer’s disease such as amyloid peptides (1-38, 1-40 and 1-42), tau protein and beclin 1, which can predict the progression of brain neurodegeneration in future. In this paper, we report for the first time the significant changes in the above molecules in the blood after asphyxia compared to healthy controls during the 1–7, 8–14 and 15+ days ELISA test.

## 1. Introduction

Perinatal asphyxia is an important cause of death in term infants, currently estimated to be responsible for approximately 23% of neonatal deaths worldwide [1,2,3]. Thus, neonatal asphyxia is a clinical problem resulting in sudden death or lifelong disability. It is associated with cerebral cessation of blood flow and impaired gas exchange during labor, resulting in multiple organ failure [2]. The most undesirable effect of asphyxia is the progressive neurodegeneration of the brain called hypoxic–ischemic encephalopathy [2]. The incidence of hypoxic–ischemic encephalopathy is estimated at 4–40/1000 live births in developing countries and 1–2/1000 live births in developed countries [2]. Long-term observation of these patients indicates the development of cerebral palsy, visual, auditory, cognitive dysfunctions, epilepsy and other serious neurological deficits [1,2,3]. Perinatal asphyxia later in life causes neurodegeneration of the brain similar to Alzheimer’s disease [4,5,6]. These data indicate that symptoms of Alzheimer’s disease can begin during perinatal asphyxia and develop into full-blown disease later in life, suggesting that asphyxia may be a causative factor in the development of Alzheimer’s disease [4,5,6].

The management of a newborn after perinatal asphyxia is very challenging due to the lack of measurable biomarkers that would be able to evaluate the severity of pathology in the future, help in the selection of treatment, to make easier assess therapy results and predict future prognosis. Currently, the assessment of changes and prognosis after perinatal asphyxia of the newborn is based on clinical symptoms, electrophysiological monitoring and imaging studies [7]. MRI for hypoxic–ischemic encephalopathy is routinely performed a few days after birth due to its lack of early sensitivity to hypoxic damage, and requires transfer of the infant from the intensive care unit to the imaging room, which is inherently rather not practiced. Both of these difficulties appear to be overcome by ultrasound of the head, but ultrasound of the head is not well studied or widely used due to its low sensitivity [8]. Biochemical assessment of the severity and extent of brain damage after perinatal asphyxia at birth is routinely assessed by cord blood gas analysis, but this test is a poor predictor of trauma sequelae [9]. In addition to brain damage, the most commonly affected organs are the heart, kidneys, lungs, liver and hematological system, which release their own specific markers into the blood, that play an important role in assessing the severity and duration of the injury and the long-term outcome [10]. As a result, new single biomarker studies are emerging that use genomic [11,12] and proteomic profiling to comprehensively understand the effects of asphyxia and treatment progress at all stages following perinatal injury [13], which will probably also indicate new therapeutic targets. Quantitative biomarkers that can detect subclinical changes at a stage where routine monitoring or brain imaging is still mute would be a major advance in the care of newborns after asphyxia [10]. Biomarkers should be able to screen newborns for post-asphyxia brain injury, monitor lesion progression and response to treatment with sequential measurements due to their short half-life and correlation with the areas of brain lesions noted by ultrasound or MRI [10]. Understanding the influence of perinatal asphyxia on the time changes in neurodegenerative biomarkers in the blood, which would be routinely used for evaluation the severity of postpartum encephalopathy, would be an important step in advancing the accuracy of predicting the consequences of brain injuries [9,14].

The development of neurodegeneration in the brain after perinatal asphyxia triggers processes at the proteomic and genomic levels, where time-dependent tracking of elements of brain-derived neurochemical traces in the blood can serve as biomarkers [14]. Identified biomarkers may improve the ability to assess the severity and progression of changes following perinatal asphyxia, as well as current treatment [14].

It should be noted that blood levels of tau protein and different amyloids with the addition of an autophagy marker following perinatal asphyxia have not been studied simultaneously. More accurate early predictive markers are urgently needed to guide clinicians when to use neuroprotective therapy. However, there are currently no neurodegenerative biomarkers that can be easily tested in readily available body fluids, such as blood, early enough to be useful in guiding treatment [15]. Using the ELISA method, our research aimed to investigate whether biomarkers of neuronal damage in the brain, such as tau protein [15,16], amyloid peptides 1-38, 1-40 and 1-42 [17] and beclin 1 appear in the blood of patients after perinatal asphyxia compared to a group of newborns without asphyxia.

## 2. Results

### 2.1. Level of the β-Amyloid Peptide 1-38 in Blood

After perinatal asphyxia, the level of β-amyloid peptide 1-38 in the 1–7 (n = 9), 8–14 (n = 15) and 15+ (n = 16) age groups was above the control (n = 16) values. Control values are the minimum 8.03 pg/mL and maximum 19.65 pg/mL with a median 14.99 pg/mL. In the 1–7 day group, the minimum was 15.80 pg/mL and the maximum 30.30 pg/mL with a median 21.12 pg/mL. In the 8–14 day group, the minimum was 18.57 pg/mL and the maximum 55.77 pg/mL with a median 26.00 pg/mL. In the group over 15 days, the minimum was 15.17 pg/mL and the maximum 32.31 pg/mL with a median 19.75 pg/mL. Figure 1 illustrates the changes in the mean levels of the β-amyloid peptide 1-38 in the blood of the studied groups. The changes between the groups were statistically significant (Kruskal–Wallis test).

### 2.2. Level of the β-Amyloid Peptide 1-40 in Blood

After perinatal asphyxia, the level of β-amyloid peptide 1-40 in the 1–7 (n = 5), 8–14 (n = 8) and 15+ (n = 8) age groups and control (n = 9) is presented in Figure 2. Control values are the minimum 89.00 pg/mL and maximum 104.90 pg/mL with a median 91.00 pg/mL. In the 1–7 day group, the minimum was 104.90 pg/mL and the maximum 104.90 pg/mL with a median 104.92 pg/mL. In the 8–14 day group, the minimum was 20.40 pg/mL and the maximum 56.00 pg/mL with a median 42.00 pg/mL. In the group over 15 days, the minimum was 64.30 pg/mL and the maximum 75.70 pg/mL with a median 73.75 pg/mL. Figure 2 illustrates the changes in the mean levels of the β-amyloid peptide 1-40 in the blood of the studied groups. Groups 8–14 and 15+ showed a significant decrease in amyloid peptide 1-40 below control values (Kruskal–Wallis test).

### 2.3. Level of the β-Amyloid Peptide 1-42 in Blood

After perinatal asphyxia, the level of β-amyloid peptide 1-42 in the 1–7 (n = 5), 8–14 (n = 8) and 15+ (n = 8) age groups and control (n = 8) is presented in Figure 3. Control values are the minimum 69.40 pg/mL and maximum 78.00 pg/mL with a median 75.50 pg/mL. In the 1–7 day group, the minimum was 39.00 pg/mL and the maximum 60.00 pg/mL with a median 54.30 pg/mL. In the 8–14 day group, the minimum was 94.60 pg/mL and the maximum 99.40 pg/mL with a median 96.95 pg/mL. In the group over 15 days, the minimum was 92.00 pg/mL and the maximum 104.90 pg/mL with a median 96.80 pg/mL. Figure 3 illustrates the changes in the mean levels of the β-amyloid peptide 1-42 in the blood of the studied groups. Groups 8–14 and 15+ showed a significant increase in amyloid peptide 1-42 above control values (Kruskal–Wallis test).

### 2.4. Level of the Tau Protein in Blood

After perinatal asphyxia, the level of tau protein in the 1–7 (n = 8), 8–14 (n = 8) and 15+ (n = 8) age groups and control (n = 8) is presented in Figure 4. Control values are the minimum 222.40 pg/mL and maximum 333.20 pg/mL with a median 268.70 pg/mL. In the 1–7 day group, the minimum was 635.00 pg/mL and the maximum 650.00 pg/mL with a median 641.18 pg/mL. In the 8–14 day group, the minimum was 117.70 pg/mL and the maximum 240.00 pg/mL with a median 222.75 pg/mL. In the group over 15 days, the minimum was 252.90 pg/mL and the maximum 329.00 pg/mL with a median 311.50 pg/mL. Figure 4 illustrates the changes in the mean levels of the tau protein in the blood of the studied groups. Group 1–7 showed a significant increase in tau protein above control values (Kruskal–Wallis test).

### 2.5. Level of the Beclin 1 in Blood

After perinatal asphyxia, the level of beclin 1 in the 1–7 (n = 5), 8–14 (n = 8) and 15+ (n = 8) age groups and control (n = 8) is presented in Figure 5. Control values are the minimum 0.00 ng/mL and maximum 0.00 ng/mL with a median 0.00 ng/mL. In the 1–7 day group, the minimum was 0.00 ng/mL and the maximum 4.00 ng/mL with a median 2.50 ng/mL. In the 8–14 day group, the minimum was 0.00 ng/mL and the maximum 1.00 ng/mL with a median 0.82 ng/mL. In the group over 15 days, the minimum was 0.20 ng/mL and the maximum 0.80 ng/mL with a median 0.65 ng/mL. Figure 5 illustrates the changes in the mean levels of the beclin 1 in the blood of the studied groups. Groups 1–7 and 8–14 showed a significant increase in beclin 1 above control values (Kruskal–Wallis test).

## 3. Discussion

The aim of this study was to determine whether changes in protein levels in the blood of newborns born after perinatal asphyxia are related to damage neurons, as in Alzheimer’s disease, being a direct reaction to asphyxia. This work builds on our previous studies in which we presented changes in the expression of genes associated with Alzheimer’s disease, such as the *amyloid protein precursor*, *β-secretase*, *presenilin 1* and *presenilin 2*, in the blood lymphocytes of neonates with untreated perinatal asphyxia or treated with hypothermia [11,12]. It is highly probable that the increase in the blood level of β-amyloid peptide 1-42 and the expression of *presenilin 1* and *2* genes in lymphocytes, starting 15 days after perinatal asphyxia, reflects the onset of progressive brain neurodegeneration and at the same time indicates the systemic nature of this process. This suggestion is also supported by a significant increase in blood tau protein levels 1–7 days after asphyxia in our study and in the hippocampus after brain ischemia, where tau protein was assessed by microdialysis [18]. This line of thought is also supported by a significant increase in blood beclin 1 level after asphyxia.

An elevated level of tau protein in the blood indicates the development of a serious pathology of the central nervous system, in particular massive pathological changes in neuronal cells, including their progressive death [16,19]. Our observations are consistent with other studies that showed an increase in tau protein levels 3–7 days post-asphyxia [14,16]. In addition, MRI studies have shown that elevated levels of tau protein on days 2–3 after hypothermia treatment are associated with death or severe neuronal damage [19]. Additionally, the level of total tau protein, which was tested by hippocampal microdialysis, was increased in experimental hypoxic–ischemic encephalopathy after hypothermic cardiac arrest [18]. Elevated blood tau protein levels have also been associated with impaired post-asphyxia brain autoregulation [19]. The results of yet another study showed an indisputable negative correlation between the level of tau protein in the blood and the developmental rate of newborns after perinatal asphyxia [20]. Study indicates that the concentration of tau protein in the blood during the first 24 h after birth can be used as a marker for the early diagnosis of neonatal hypoxic–ischemic encephalopathy and for predicting poor neurodevelopmental outcomes [20]. Another study showed that elevated levels of tau protein in the blood correlated with the severity of injury and was associated with poorer neurodevelopmental outcome over 1 year [21]. Our data, as well as data from other studies, indicate that blood tau protein levels measured in the first week of life after asphyxia may predict neurodevelopmental outcomes later in life [16,20,21].

This study showed for the first time the involvement of beclin 1 autophagy protein in the development of post-asphyxia brain pathology in the first three weeks of life. Elevated blood levels of beclin 1, a protein associated with autophagy, coincided temporally with titers of tau protein in our study. A massive increase in beclin 1 levels was observed in the first week after asphyxia, which was in response to the behavior of the tau protein in this brain pathology. In the following weeks of observation, the level of these proteins steadily decreased to control values, indicating that they were associated with the body’s acute reaction to the sudden pathology.

Changes in beclin 1 and tau protein levels in the blood preceded the increase in changes in amyloid peptides 1-38 and 1-42. Studies of amyloid levels in the hippocampus after ischemia by microdialysis have shown that an acute ischemic episode can trigger peptide production via the BACE1 pathway as early as 4 to 6 h of reperfusion [22]. Blood levels of amyloid peptide 1-40 in our study oscillated within control values after asphyxia. While levels of amyloid peptide 1-38 were significantly elevated during the three weeks of observation. In contrast, blood levels of amyloid peptide 1-42 were significantly elevated in the second and third week of follow-up after asphyxia. From our observations, in the second and third week after asphyxia, there was a decrease in the ratio of amyloid peptide 1-40 to 1-42, similar to Alzheimer’s disease pathology [23]. Indeed, amyloid peptide 1-42 is the main component of amyloid plaques and is shown to be neurotoxic [23]. Therefore, amyloid peptide 1-42 is thought to be a key player in initiating amyloid plaque formation [23]. It seems that following perinatal asphyxia, amyloid protein precursor metabolism is dysregulated and may contribute to the development of Alzheimer’s disease in the future by increasing overall amyloid production and reducing the amyloid peptide 1-40/1-42 ratio [23,24,25]. Amyloid can induce tau protein hyperphosphorylation through the activation after asphyxia tau protein kinase GSK-3β [26,27,28]. In addition, GSK-3β has been shown to catalyze the phosphorylation of tau protein through elevated α-synuclein as a result of ischemia, which is important for secondary brain damage after ischemia [27]. Additionally, amyloid induction neuroinflammation may also contribute to tau protein pathology [23]. Amyloid plays a primary role in activating several innate immune pathways, causing neuroinflammatory response and releasing inflammatory cytokines, such as interleukin-1β [23]. Conversely, increasing interleukin-1β signaling pathway was shown to exacerbate tau protein pathology [23]. Abundant evidence indicates that amyloid-induced neurotoxicity occurs in a tau protein-dependent manner in the post-ischemic and Alzheimer’s disease brain, especially in the hippocampus, which is primarily responsible for memory [23,24,25] and probably this phenomenon is dominant post-asphyxia. Building on previous observations suggesting that post-asphyxia hypoxic–ischemic encephalopathy may be associated later in life with a clinical diagnosis of Alzheimer’s disease, current evidence suggests that it may be related to increased amyloid and tau protein load in the brain. Perinatal asphyxia and Alzheimer’s disease are characterized by a drastic reduction in cerebral blood flow, resulting in an increase in amyloid production and pathological modification of the tau protein. Thus, the relationship between brain neurodegeneration after asphyxia and the clinical diagnosis of Alzheimer’s disease later in life can be explained by coexisting ischemic brain injury. Therefore, early, subtle cerebral vascular damage does not preclude meeting the clinical criteria for Alzheimer’s disease in the future. This means that the initial association between perinatal asphyxia and Alzheimer’s disease may not initially reflect Alzheimer’s disease itself, but rather an Alzheimer’s disease-type dementia, which will certainly end up with full-blown Alzheimer’s disease over the years. Our data show association between perinatal asphyxia neuronal/brain damage and Alzheimer’s disease-specific factors such as amyloid and tau protein. Data analysis also indicates that elevated levels of tau protein, amyloid peptides and beclin 1 in the blood are a predictors of future neurodevelopmental disorders.

Finally, studies have shown that the levels of the molecules tested during 3 weeks and more after an episode of asphyxia suggest the possibility of reflecting pathological processes in the damaged brain, which indicate the systemic nature of the disease. Based on these preliminary studies, the levels of the above molecules as possible markers of brain neurodegeneration, as well as their potential as markers of brain neurodegeneration due to asphyxia, should be seriously considered. It should be emphasized here that, despite these efforts, the understanding and neurological prognosis of post-suffocation hypoxic–ischemic encephalopathy is still limited. There is a great need for early and accurate prognostic methods, both to avoid long-term treatment of patients who fail further treatment and to ensure optimal treatment for patients with potential for recovery. Taken together, our data suggest that blood levels of amyloids, tau protein and beclin 1 may be promising biomarkers for predicting the extent of brain neurodegeneration and the neurodevelopmental effects of asphyxia in later life.

It should be noted that our study has several limitations. First, we were limited by the volume of blood available for analysis, which made it impossible to evaluate all bi-omarkers in the same patient at the same time. The use of virtually residual blood samples in the research meant that we could not have access to blood tests at a time of our choos-ing. This meant that we had to gather material in seven-day subgroups. This approach resulted in different sample freezing times. Our results were also influenced by the small number of research groups and their size and time of observation. In addition, we were unable to assess the temporal relationship between the biomarkers studied and clinical assessments because we did not always have a complete clinical picture from other clinical studies such as MRI or a full neurological assessment.

Therefore, the usefulness of the proposed biomarkers in predicting the consequences of asphyxia requires further research. Understanding the effect of asphyxia on the temporal trajectory of serum biomarkers to determine its severity is an important step in increasing the accuracy of post-asphyxia brain injury diagnosis and building predictive models. Future studies are warranted to confirm or disprove the value in clinical practice of the above research proposal. Despite these limitations, we noted that the biomarkers under study may be clinically useful biomarkers for identifying neonates at risk of secondary brain injury due to asphyxia. Thus, validation of the biomarkers being studied as putative biomarkers on a large scale is warranted. Including studies using MRI to evaluate amyloid and tau protein in the brain and their correlation with proposed blood biomarkers after long-term follow-up after asphyxia.

## 4. Materials and Methods

In this analysis, we used children with a history of perinatal asphyxia during postnatal follow-up 15 days and more. Newborns belonged to the strictly defined area of care and hospitalization of the Department of Neonatal and Infant Pathology, Medical University of Lublin, Poland. The criteria of the principles confirming the history of asphyxia were determined by the following parameters:Newborns (preterm and full-term) > 31 weeks of gestational age.Acidosis with pH < 7.0 (in umbilical cord or blood sample obtained throughout the 60 min post-asphyxia).Or base deficit > −12.Or Apgar score of 0–5 at 10 min after birth or during 10 min of resuscitation.The presence of multiple organ dysfunctions.Clinical manifestations of encephalopathy: aberrant oculomotor or pupillary activity, feeble or elusive suck, episodic breathing/apnea, or seizures.Neurological deficits cannot be related to alternative disease.

The research procedure was accepted by the Ethics Committee of the Medical University of Lublin (25 April 2013, consent no. KE-0254/118/2013). Then, after getting the written permission of all neonatal caregivers, blood for research was collected simultaneously with blood samples for routine laboratory tests for further diagnosis and/or treatment.

### 4.1. Research Groups

The research included 40 neonates with no perinatal complications (control group) and 40 neonates following perinatal asphyxia (research group). The control and research group were split into three different age subgroups 1–7 days, 8–14 days and 15+ days. In the control and research group, the number of newborns used is presented in the results and in Figure 1, Figure 2, Figure 3, Figure 4 and Figure 5. Children in control group were hospitalized due to breast milk jaundice, allergic rash, infantile colic, poor weight gain and respiratory infections. Control neonates were born following uncomplicated delivery and without life-threatening health complications. Small patients from the control and study group were simultaneously hospitalized in the Department of Neonate and Infant Pathology, Medical University of Lublin. The purpose of hospitalization of the control group was to assess the general development of the newborn and/or to establish the diagnosis and treatment of the above-mentioned symptoms. General clinical characteristics of neonates used in this study are presented in Table 1.

### 4.2. Elisa Studies

Serum concentrations of β-amyloid peptides (1-38, 1-40, 1-42), tau protein, and beclin 1 were detected using the following ELISA kits: 1. Amyloid 1-38 High Sensitive ELISA, IBL International GmbH, Humburg, Germany (Catalog # JP27717). 2. Amyloid 1-40 Human ELISA Kit, Thermofisher Scientific Invitrogen (Cat # KHB3481), Waltham, MA, USA. 3. Amyloid 1-42 Human ELISA Kit, Ultrasensitive, Thermofisher Scientific Invitrogen, USA (Cat # KHB3544). 4. Tau protein (Total) Human ELISA Kit, Thermofisher Scientific Invitrogen, USA. (Cat # KHB0041). 5. ELISA Kit for Beclin 1 (BECLIN 1), Cloud-Clone Corp., Katy, TX, USA (Cat#SEJ557).

From each patient, fasting blood samples (~5 mL) were collected, the plasma was centrifuged and frozen at −70 °C. Then each assay was performed according to the manufacturer’s instructions, and the optical density value was measured using a spectrophotometer (ETIMax 3000, DiaSorin, Saluggia, Italy) at 450 nm. The experiments were performed in triplicate and repeated at least once. Frozen plasma samples were slowly thawed, and a 20 μL aliquot was diluted 1:10 with undiluted standard dilution buffer provided with each ELISA Kit. A 50 μL aliquot of this solution was added to the appropriate well (according each assay).

All reagents used were supplied in the kit including the standards dilution buffer. Wash buffer was prepared by diluting the supplied concentrate 1:25 with purified water. Standards were prepared by diluting the supplied standard with standard dilution buffer as specified on labeling to 2000 pg/mL. Serial dilutions were made into standard dilution buffer to 1000, 500, 250, 125, 62.50, and 31.25 pg/mL or ng/mL. Volumes of 100 μL of standards were added to wells in duplicate. A volume of 50 μL of standard dilution buffer was added to the rest of the wells. The ELISA plate was covered and incubated for 2 h at room temperature on a rotary shaker (1200 rpm) and thereafter washed 4 times. A 100 μL aliquot of anti-protein detection specific antibody was added to each well, and the plate was incubated as above for 1 h and thereafter washed 4 times. Antirabbit HRP concentrate was diluted 100× into HRP diluent and added to wells at 100 μL/well, and the plate is incubated for 30 min. After emptying the plate 100 μL of stabilized chromogen was added to each well, and the uncovered plate was incubated at room temperature in the dark for 20–30 min on the rotary shaker. The reaction was completed by addition of 100 μL of provided stop solution, and the plate was then read at 450 nm in a plate reader. Reader software (32-bit Microsoft Windows 2000) calculated standard curve, concentration, SD, and CV% (4 parameter algorithm).

### 4.3. Statistical Analysis of the Results

Statistical evaluation of the results was carried out using the Statistica v. 13.3 software (Tibco Corporation, Palo Alto, CA, USA) with the help of non-parametric Kruskal–Wallis test with the “z” test-multiple analyses of differences between groups. Data are presented as the means ± standard deviation (SD). Statistical significance was adopted at *p* ≤ 0.05.

## Figures and Tables

**Figure 1 ijms-24-13292-f001:**
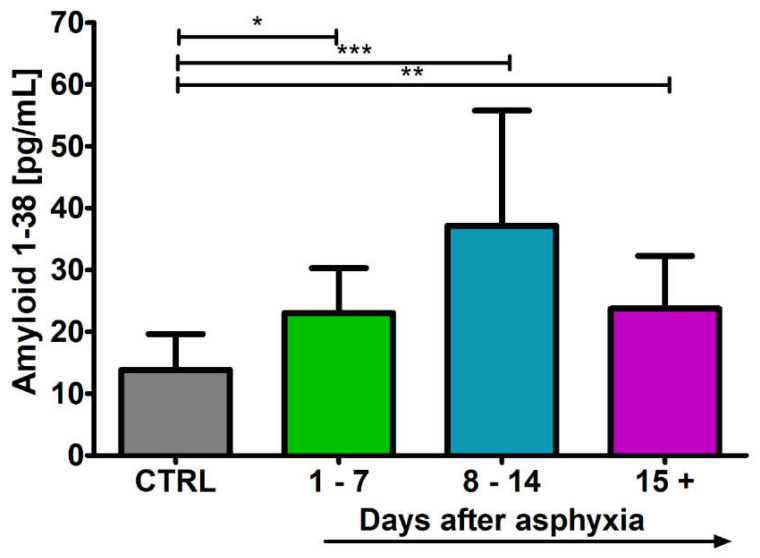
Mean levels of β-amyloid peptide 1-38 in plasma in control (CTRL) (n = 16) and following asphyxia in age groups 1–7 (n = 9), 8–14 (n = 15), and 15+ (n = 16) days. Marked SD, standard deviation. N-number of patients. The indicated statistically significant differences in the level of peptide between control and 1–7 (z = 2.897, *p* = 0.0226), 8–14 (z = 4.710, *p* = 0.00002) and 15+ days (z = 3.246, *p* = 0.007) after asphyxia (Kruskal–Wallis test). * *p* ≤ 0.05, ** *p* ≤ 0.01, *** *p* ≤ 0.001.

**Figure 2 ijms-24-13292-f002:**
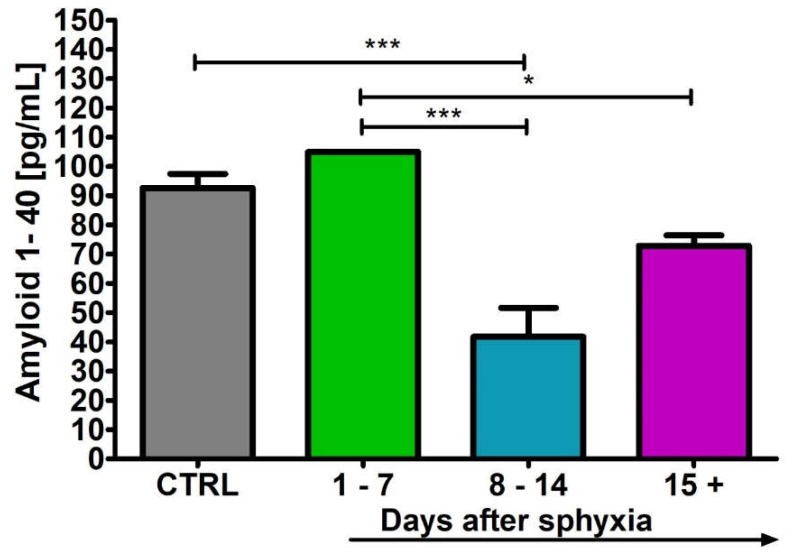
Mean levels of β-amyloid peptide 1-40 in plasma in control (CTRL) (n = 9) and following asphyxia in age groups 1–7 (n = 5), 8–14 (n = 8), and 15+ (n = 8) days. Marked SD, standard deviation. N-number of patients. The indicated statistically significant differences in the level of peptide between control and 8–14 (z = 3.922, *p* = 0.0005), 1–7 and 8–14 (z = 4.583, *p* = 0.00003) and 1–7 and 15+ days (z = 2.989, *p* = 0.02) after asphyxia (Kruskal–Wallis test). * *p* ≤ 0.05, *** *p* ≤ 0.001.

**Figure 3 ijms-24-13292-f003:**
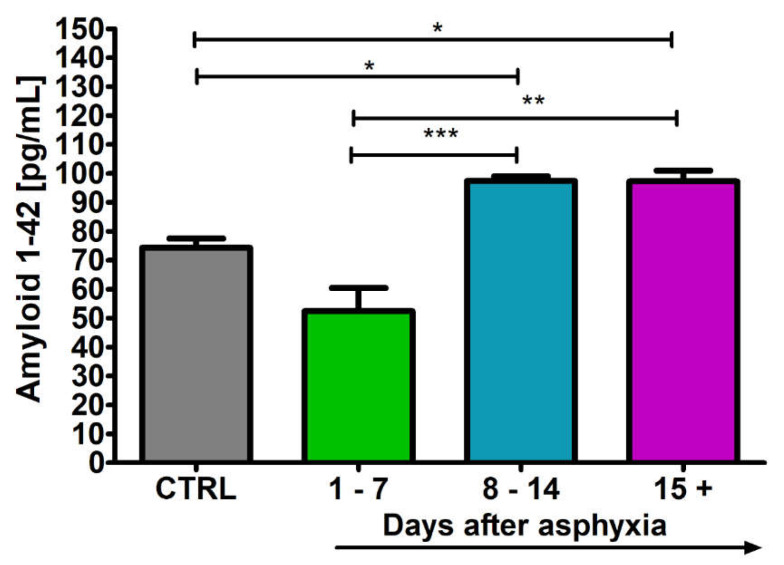
Mean levels of β-amyloid peptide 1-42 in plasma in control (CTRL) (n = 8) and following asphyxia in age groups 1–7 (n = 5), 8–14 (n = 8), and 15+ (n = 8) days. Marked SD, standard deviation. N-number of patients. The indicated statistically significant differences in the level of peptide between control and 8–14 (z = 2.995, *p* = 0.02) and 15+ (z = 2.643, *p* = 0.049), 1–7 and 8–14 (z = 3.966, *p* = 0.0004) and 1–7 and 15+ days (z = 3.657, *p* = 0.002) after asphyxia (Kruskal–Wallis test). * *p* ≤ 0.05, ** *p* ≤ 0.01, *** *p* ≤ 0.001.

**Figure 4 ijms-24-13292-f004:**
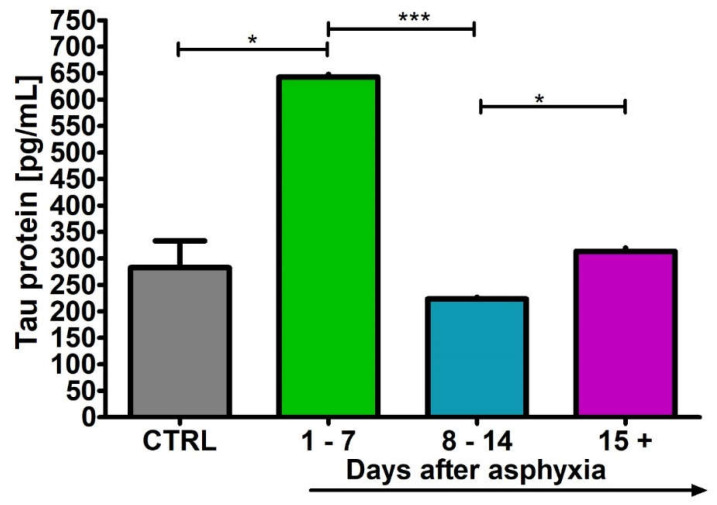
Mean levels of tau protein in plasma in control (CTRL) (n = 8) and following asphyxia in age groups 1–7 (n = 8), 8–14 (n = 8), and 15+ (n = 8) days. Marked SD, standard deviation. N-number of patients. The indicated statistically significant differences in the level of protein between control and 1–7 (z = 2.772, *p* = 0.03), 1–7 and 8–14 (z = 5.116, *p* = 0.000002) and 8–14 and 15+ days (z = 2.989, *p* = 0.03) after asphyxia (Kruskal–Wallis test). * *p* ≤ 0.05, *** *p* ≤ 0.001.

**Figure 5 ijms-24-13292-f005:**
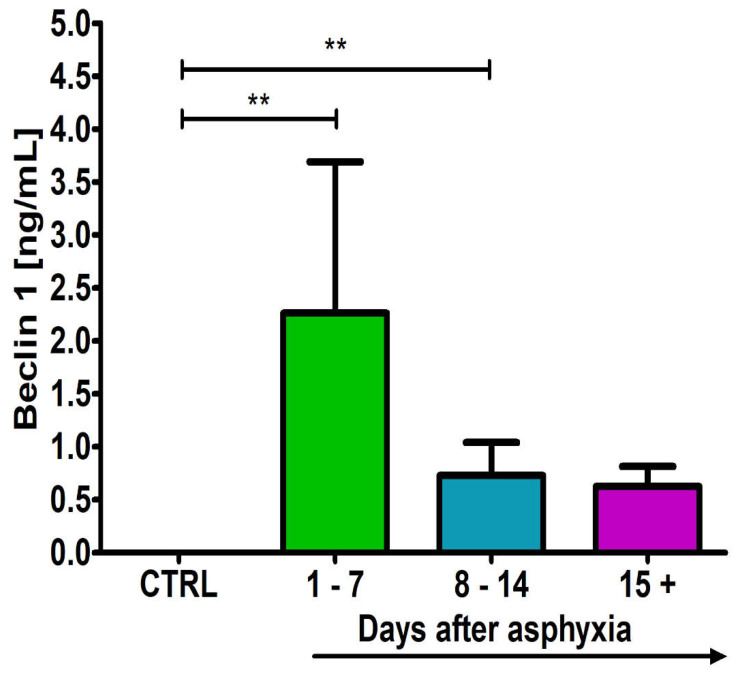
Mean levels of beclin 1 (autophagy protein) in plasma in control (CTRL) (n = 8) and following asphyxia in age groups 1–7 (n = 5), 8–14 (n = 8), and 15+ (n = 8) days. Marked SD, standard deviation. N-number of patients. The indicated statistically significant differences in the level of protein between control and 1–7 (z = 3.626, *p* = 0.002) and 8–14 (z = 3.215, *p* = 0.008) after asphyxia (Kruskal–Wallis test). ** *p* ≤ 0.01.

**Table 1 ijms-24-13292-t001:** Clinical characteristics of included neonates.

VARIABLES	N	CONTROL		ASPHYXIA	
		Mean ± SD	Median	Mean ± SD	Median
**Age** (Days)	10	15 ± 7	15	10 ± 7	7
**Gestational Age** (Weeks)	10	39 ± 1	39	39 ± 2	39
**Apgar Score** (1 min)	10	10 ± 0	10	2 ± 1.5	1.50
**WBC** (/µL)	10	13,640 ± 2651	13,870	25,298 ± 9300	24,300
**RBC** (×1000/µL)	10	4654 ± 539	4855	4498 ± 761	4715
**Hct** (%)	10	44 ± 5	46	48 ± 10	49
**PLT** (×1000/µL)	10	446 ± 165	455	250 ± 64	260
**Lymphocytes** (/µL)	10	7180 ± 907	7520	5609 ± 2247	4640
**pH**	10	7.41 ± 0.02	7.41	7.23 ± 0.25	7.23
**BE** (mmol/L)	10	0.80 ± 1.7	−0.10	−12.77 ± 9.9	−11.20
**Birth Weight** (g)	10	3572 ± 664	3675	3572 ± 664	3620

RBC—red blood cells, WBC—white blood cells, PLT—platelets, Hct—hematocrit, BE—base deficit, and N—number of patients.

## Data Availability

The data presented in this study are available on request from the corresponding author.

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
