# Peer review of "Preservation of Biomarkers Associated with Alzheimer’s Disease (Amyloid Peptides 1-38, 1-40, 1-42, Tau Protein, Beclin 1) in the Blood of Neonates after Perinatal Asphyxia"

_ijms, 2023, doi:10.3390/ijms241713292_

Round 1

Reviewer 1 Report

Preservation of biomarkers associated with Alzheimer's disease 2 (amyloid peptides 1-38, 1-40, 1-42, tau protein, beclin 1) in the 3 blood of neonates after perinatal asphyxia

The study aims to demonstrate the changes in serum biomarkers of neuronal injury characteristic for Alzheimer’s disease to predict the progression of brain neurodegeneration in future. The approach and the overall design of the study are good.  The manuscript is well written. Include a separate subsection in materials and methods describing the statistical analysis performed. Correct the abbreviation as ‘ELISA’. The circulating levels of various miRNAs that are involved in neurodegenerative diseases are suggested for future studies. Include the limitations and future implications of the study.

Author Response

Review 1.

All changes are in red.

Preservation of biomarkers associated with Alzheimer's disease (amyloid peptides 1-38, 1-40, 1-42, tau protein, beclin 1) in the blood of neonates after perinatal asphyxia

The study aims to demonstrate the changes in serum biomarkers of neuronal injury characteristic for Alzheimer’s disease to predict the progression of brain neurodegeneration in future. The approach and the overall design of the study are good.  The manuscript is well written.

Thanks.

Include a separate subsection in materials and methods describing the statistical analysis.

Done

Correct the abbreviation as ‘ELISA’.

Corrected.

The circulating levels of various miRNAs that are involved in neurodegenerative diseases are suggested for future studies.

We will consider this proposal in future research.

Include the limitations and future implications of the study.

Done.

In addition, we found a typo in Figure 5 and corrected "mg" to "ng".

Reviewer 2 Report

I appreciate the author presenting this clinical application useful research article. My comments are as follow:

The authors propose new biochemical markers after neonatal asphyxia. Current data can only show statistically significant differences in some biochemical performance at different time points after neonatal asphyxia compared to normal newborns. If the authors could correlate the severity of the disease with these indicators they might be able to concluded as the conclusion mentioned “our data indicate that levels of amyloids, tau protein and beclin 1 may be promising biomarkers for predicting the severity of post-asphyxia brain neurodegeneration”

Author Response

Review 2.

All changes are in red.

I appreciate the author presenting this clinical application useful research article.

Thanks.

My comments are as follow:

The authors propose new biochemical markers after neonatal asphyxia. Current data can only show statistically significant differences in some biochemical performance at different time points after neonatal asphyxia compared to normal newborns. If the authors could correlate the severity of the disease with these indicators they might be able to concluded as the conclusion mentioned “our data indicate that levels of amyloids, tau protein and beclin 1 may be promising biomarkers for predicting the severity of post-asphyxia brain neurodegeneration”.

We have toned down the wording of our final conclusion. In future studies, we will try to relate the severity of the disease to the indicators studied. In addition, in the discussion, we presented the limitations of the research and their future implications.

In addition, we found a typo in Figure 5 and corrected "mg" to "ng".

Round 2

Reviewer 2 Report

The authors have replied my comments item-by-item. I have nor more questions. Accept is my final decision.